



# Modeling the Influence of Chain Length on SOA Formation via Multiphase Reactions of Alkanes

Azad Madhu[1], Myoseon Jang[1], David Deacon[1]

[1]Engineering School of Sustainable Infrastructure and Environment, University of Florida, Gainesville, 32608, United States of America

*Correspondence to*: Myoseon Jang (mjang@ufl.edu)

**Abstract.** Secondary Organic Aerosol (SOA) from diesel fuel is known to be significantly sourced from the atmospheric oxidation of aliphatic hydrocarbons. In this study, the formation of linear-alkane SOA was predicted using the Unified Partitioning Aerosol Phase Reaction (UNIPAR) model that simulated multiphase reactions of hydrocarbons. In the model, the formation of oxygenated products from the photooxidation of linear alkanes was simulated using a near-explicit gas kinetic mechanism. Autoxidation paths integrated with alkyl peroxy radicals were added to the Master Chemical Mechanismv3.3.1 to improve the formation of low volatility products in the gas phase and better predict SOA mass. The resulting gas products were then classified into volatility-reactivity based lumping groups that are linked to mass-based stoichiometric coefficients. The SOA mass in the UNIPAR model is produced via three major pathways: partitioning of gaseous oxidized products onto both the organic and wet inorganic phases; oligomerization in organic phase; and reactions in the wet inorganic phase (acid-catalyzed oligomerization and organosulfate formation). The model performance was demonstrated for SOA data that were produced through the photooxidation of a homologous series of linear alkanes ranging from C9 to C15 under varying environments (NOx levels, temperature, and inorganic seed conditions) in a large outdoor photochemical smog chamber. The product distributions of linear alkanes were mathematically predicted as a function of carbon number using an incremental volatility coefficient (IVC) to cover a wide range of alkane lengths. The prediction of alkane SOA using the incremental volatility-based product distributions, which were obtained with C9-C12 alkanes, was evaluated for prediction of C13 and C15 chamber data and further extrapolated to predict the SOA from longer chain alkanes (□C15) that can be found in diesel. The model simulation of linear alkanes in diesel fuel suggests that SOA mass is mainly produced by alkanes C15 and higher. Alkane SOA is insignificantly impacted by the reactions of organic species in the wet inorganic phase due to the hydrophobicity of products, but significantly influenced by gas-particle partitioning.

## 1 Introduction

Secondary organic aerosol (SOA) represents a significant proportion (20%-90%) of fine particulate matter in ambient air (Kanakidou et al., 2005). SOA is formed from the atmospheric photooxidation of hydrocarbon (HC) species and can have significant impacts on climate, and human health (U.S. EPA 2019; Nel, 2005; Shrivastava et al., 2017; World Health, 2016).



To properly assess SOA burdens, especially in urban areas, precursor alkane species must be accounted for. Alkanes are released into the atmosphere primarily through varying anthropogenic sources such as fossil fuels and volatile chemical products (i.e. personal care products, paints, pesticides, etc.) (Li et al., 2022; Wu et al., 2019). Alkanes have been shown to be significant components of gasoline exhaust (6-18%) and diesel exhaust (18-31%) and vehicle emissions are the main

source in urban areas (Caplain et al., 2006). Whereas the majority of SOA produced from gasoline sources is from its aromatic content, about half of the SOA produced from diesel can be attributed to aliphatic compounds (Gentner et al., 2012). Currently, long chain alkanes are recognized to be important precursors of SOA production with increasing carbon numbers in linear alkanes leading to higher SOA yields (Aumont et al., 2012). Branched and cyclic alkanes will tend to have different yields compared to linear alkanes with the same carbon number. This is partially because branched alkanes tend to

produce more volatile products and cyclic alkanes produce less volatile products relative to linear alkanes (Lim and Ziemann, 2009).

Previous studies that model SOA formation have often had a discrepancy between predicted SOA formation and field observations (Shrivastava et al., 2017; Volkamer et al., 2006). This discrepancy can be explained by several reasons. Firstly, the contribution of intermediate volatility organic compounds (IVOCs) have often been ignored during model simulations

(Robinson Allen et al., 2007). Multiple studies in recent years have attempted to correct this discrepancy by including IVOCs. Hodzic et al. (2010) found that the inclusion of IVOCs led to a substantial improvement in model predictions of SOA compared to data collected from the MILAGRO field experiment. Alkanes are important IVOCs in urban environments. Lee-Taylor et al. (2011) attempted to simulate the same MILAGRO field data using a near-explicit gas mechanism (GECKO-A) to determine SOA mass based on gas-particle partitioning of possible intermediate products. They

predicted the field data well and they found that more that 75% of SOA produced in their model were from products of larger linear alkane species.

Secondly, currently existing gas mechanisms for precursor hydrocarbons have a large level of uncertainty due to unidentified oxidation paths. For example, Xavier et al. (2019) found that the formation of terpene SOA simulated by using exclusively the MCM mechanism consistently underpredicted experimental values but the inclusion of peroxy radical autoxidation

mechanisms (PRAM) improved predictions by increasing the formation of low volatility products. Like terpenes, autoxidation occurring on the products of alkane species can produce low volatility products that readily partition to the particle phase, but it has not been included in the current explicit mechanism. The addition of those reaction pathways can improve currently existing mechanisms and consequently, SOA predictions.

Thirdly, relying on an improvement of the gas mechanism alone is not sufficient for improving predictions of SOA

production. Lee-Taylor et al. notes that their reasonable agreement for SOA predictions with field data is not necessarily realistic as they fail to consider alternative pathways for SOA formation such as particle-phase reactions. Whereas the particle-phase reactions of alkane SOA are limited, the organic matter produced by aerosol phase reactions of reactive organic compounds, originating from other precursors, can increase the partitioning of alkane oxidation products. Yang et al. (2019) simulated SOA formation in China using the nested air quality prediction modeling system (NAQPMS) but have a



similar issue to Hodzic et al. (2010) as they also exclusively use volatility-based lumping and fail to consider particle-phase reactions. Pye and Pouliot (2012) consider particle phase reactions in their simulation of regional SOA formation using the Community Multiscale Air Quality (CMAQ) model. However, the complexity of particle phase reactions in their model is limited as they use a uniform reaction rate constant for the particle phase reactions of semi-volatile SOA. Alkane SOA may not be significantly influenced by particle phase reactions compared to other precursor, but few studies have attempted to

account particle phase reactions to form alkane SOA.  For example, Cappa et al. (2013) attempts to implicitly capture particle-phase reactions through fitting model parameters to alkane SOA experimental measurements instead of explicitly modeled particle-phase reactions. Similarly, in the model derived by Zhang and Seinfeld (2013), the reactions are controlled by rate constants that are fit to experimental measurements, rather than derived from the properties of explicit or lumped compounds from alkane oxidation.

The UNIPAR model was previously developed to improve predictions by considering particle-phase reactions as well as gas-particle partitioning as mechanisms for SOA production (Beardsley and Jang, 2016; Im et al., 2014; Zhou et al., 2019). In this study, autoxidation pathways of alkyl peroxy radicals are added to currently existing alkane MCM oxidation mechanisms to include low volatility oxidation products and applied to the UNIPAR model for SOA prediction. The simulation results are compared to data collected in the UF Atmospheric PHotochemical Outdoor Reactor (UF-APHOR)

chamber under the controlled environments (NOx levels, and seed conditions). In order to simulate SOA formation from a wide range of alkane length, a mathematical equation to construct product distributions was equipped in the UNIPAR model by using explicit products of C9-C12 linear alkanes. This equation is extrapolated to predict product distributions from larger alkanes (C13 & C15), for which explicit mechanisms are unavailable in MCM, and applied to SOA prediction. These predictions are also compared to chamber data obtained for C13 and C15. Extrapolation to alkanes larger than C15 is

explored and used to project SOA formation from linear alkane species found in diesel fuel. In order to increase the feasibility of the model, UNIPAR was also integrated with the typical ozone mechanisms such as the carbon bond mechanism (Emery et al., 2015).

## 2 Experimental section

Alkane SOA was produced from photooxidation of a homologous series of linear alkanes (C9-C13, C15) using the

UF-APHOR dual chamber (52 m$^3$ each) located at the University of Florida. The detailed description of the operation of the large outdoor smog chamber can be found in a previous study (Im et al., 2014). The detailed experimental conditions of outdoor chamber experiments are summarized in Table 1. Generally, the precursor alkane HCs were evaporated through heating to introduce them into the chamber. A non-reactive gas, CCl$_4$ (Sigma-Aldrich; ≥99.5%), was introduced into the chamber as an indicator for chamber dilution. HONO was added to the chamber as a source of hydroxyl radicals. HCs,

HONO, and NO (2% in N$_2$, Airgas Inc., USA) were introduced into the smog chamber before sunrise. The HCs were studied

under two different $NO_x$ levels (high $NO_x$: $HC/NO_x$ <5 ppbC/ppb; low $NO_x$: $HC/NO_x$ >10 ppbC/ppb) and three different seed conditions (without seed; sulfuric acid, SA; and ammonium sulfate, AS). To study the impact of aerosol liquid water content (LWC) on SOA formation, experiments with Ammonium Sulfate seed were performed under dry conditions (RH was controlled less than the efflorescence RH (ERH) of AS seed) and wet conditions (RH was maintained higher than 50% to

prevent seed crystallization).

A GC-FID (7820A, Agilent Technologies, Inc., USA) was used to measure the concentration of gas-phase HCs and $CCl_4$. A photometric ozone analyzer (400E, Teledyne Technologies, Inc., USA) and a chemiluminescence $NO/NO_x$ analyzer (T201, Teledyne Technologies, Inc., USA) were used to measure the concentrations of ozone and $NO_x$ within the chamber, respectively. A Particle into liquid sampler (ADISO 2081, Applikon Inc., USA) integrated with Ion Chromatography

(Compact IC 761, Metrohm Inc., Switzerland) (PILS-IC) was used in experiments performed with inorganic seed to measure the inorganic ion concentration within the chamber. A scanning mobility particle sizer (SMPS 3080, TSI Inc., USA) was used to measure the size distribution of particles within the chamber. Previous studies that have measured the density of alkane SOA have found a range from 1 to 1.4 $g/cm^3$ (Li et al., 2020; Li et al., 2022; Lim and Ziemann, 2009; Loza et al., 2014). Aerosols from each alkane experiment in this study were assumed to have a density of 1.2 $g/cm^3$. A hygrometer

(CR1000 measurement and control system, Campbell Scientific Inc., USA) was used to measure meteorological factors (temperature, relative humidity (RH) and an ultraviolet radiometer (TUVR, Eppley Laboratory Inc., USA) was used to measure sunlight intensity. An organic carbon/elemental carbon analyzer (OC/EC model 4, Sunset Laboratory Inc., USA) was used to measure the concentration of organic carbon in aerosol every 50 minutes. The concentration of organic matter in aerosol (OM, $\mu g\ m^{-3}$) was then calculated based on the OC concentration predicted by the UNIPAR model and an OM to OC

ratio. The OM to OC ratio of SOA from alkane species decreased as the chain length increased. The concentrations of OM measured from the chamber were corrected for chamber dilution using a dilution factor and for particle wall loss to the chamber wall using a particle loss factor. An aerosol chemical speciation monitor (ACSM, Aerodyne Research Inc., USA) was used to measure the aerosol composition (sulfate, nitrate, ammonium, and OM). The compositions obtained from the ACSM were compared with measurements from the OC/EC and the PILS-IC. SOA yields ($Y$) were then calculated as the

final measured concentration of OM divided by the total consumption of HC precursors.



## 3 Model description

The UNIPAR model simulates the formation of SOA from multiphase reactions of linear alkanes including gas (g), organic (or) and inorganic (in) phases. The model utilizes a near-explicit gas phase mechanisms for the photooxidation of each alkane. The model employed the Master Chemical Mechanism (MCM v3.3.1) and gas autoxidation pathways to yield highly

oxidized molecules at given alkane and meteorological conditions under various NOx levels (HC ppbC/NOx ppb= 2□50). The resulting products were then classified into an array of 51 different lumping species based on their volatility and reactivity. Previous studies have detailed description of the lumping criteria and the mass-based stoichiometric coefficient ($\alpha_i$) of lumping group i as a function of NOx levels and the degree of aging (Zhou et al., 2019). In order to cover alkanes in a variety of carbon length, the product distributions of linear alkanes were mathematically represented as a function of carbon

number using an incremental volatility coefficient (IVC). The SOA mass in the UNIPAR model forms via three pathways: organic matter produced via partitioning of gaseous products onto both or and in phases (OMP); oligomerization in the or phase; and reactions in the wet in phase (acid-catalyzed oligomerization and organosulfate (OS) formation). The aerosol reactions of organic species form OMAR. The inputs to the UNIPAR model include the equations for $\alpha_i$; physicochemical parameters of lumping species; the consumption of HC ($\Delta$HC); the concentrations of alkyl peroxide radical (RO2) and

hydroperoxyl radical (HO2); concentrations of ionic species (sulfate and ammonium ions); temperature, and humidity. Fig. 1 illustrates the UNIPAR model frame.

### 3.1 Gas mechanisms

        Unlike the formation of highly oxidized molecules in terpene that starts with ozonolysis, the process in alkanes

begins with the reaction with an OH radical, followed by the addition of $O_2$ to form peroxyl radicals. This addition can create specific structures in some compounds which have been identified in previous literature to be capable of undergoing autoxidation reactions (Bianchi et al., 2019; Pye et al., 2019; Roldin et al., 2019; Xavier et al., 2019). An example of an autoxidation pathway added to the MCM mechanism for a linear alkane is shown in Fig. S1. Most autoxidation products from the alkanes higher than C9 belong to the low volatility group of the UNIPAR model and these products can increase

SOA mass via gas-particle partitioning. As the size of the linear alkane increases, there is an increased likelihood that some of the precursor HC is lost to the chamber wall. The rate at which alkane species partition to the wall of the UF-APHOR chamber was measured using a night-time experiment (Table S1).



### 3.2 Lumping and aging

The MCM with alkyl peroxy radical autoxidation reactions is used to simulate the gas-phase oxidation of given HC at various $NO_x$ levels (HC ppbC/$NO_x$ ppb = 2~50). A standard meteorological profile (14 June 2018) of a day with a clear sky near summer solstice was used and all simulations began before sunrise. The resulting oxidized products are lumped into an array of 51 different species. The array consists of 6 different reactivity levels (very fast, fast, medium, slow, partitioning only, and multi-alcohol) and 8 different volatility levels based on vapor pressure, along with 3 explicit species that are

lumped separately (glyoxal, methylglyoxal, and epoxydiols). For each $NO_x$ level, the lumped product distribution is represented as an array of mass-based stoichiometric coefficients ($\alpha_i$) where $i$ represents a specific lumping species. To incorporate the impact of aging on SOA formation, the dynamic $\alpha_i$ is reconstructed using two different gas phase oxidation compositions: fresh and highly oxidized. A weighted aging factor is used to dynamically change $\alpha_i$ values based on these two compositions. The aging factor ($f_a$) at a time $t$, as detailed in Zhou et al. (2019), is defined as:

$$f_a(t) = \log \frac{[HO_2] + [RO_2]}{[HC]_0} \tag{1}$$

where $[HO_2]$ and $[RO_2]$ and $[HC]_0$ are the concentrations of hydroperoxide radical, organic peroxyl radical, and initial HC, respectively. Both fresh and highly oxidized compositions are calculated for each $NO_x$ level as well as the respective aging factors. $f_a(t)$ is also converted into an aging scale that ranges from 0 (fresh composition) to 1 (highly oxidized composition) as follows (Zhou et al., 2019):

$$f_a'(t) = \frac{f_a(highly\ oxidized) - f_a(t)}{f_a(highly\ oxidized) - f_a(fresh)} \tag{2}$$

$f_a(t)$ is calculated for a given $NO_x$ level and used to dynamically calculate the $\alpha_i$ values for that same $NO_x$ level as follows (Zhou et al., 2019):

$$\alpha_i = (1 - f_a'(t))(fresh\ \alpha_i) + (f_a'(t))(highly\ oxidized\ \alpha_i) \tag{3}$$

The dynamic updates of physicochemical parameters (molecular weight (MW), O:C ratio, and hydrogen bonding (HB)) of

170 lumping species are also performed in a similar way. Further details about lumping criteria and dynamic updates of physicochemical parameters can be found in the study by Zhou et al. (2019).





### 3.3 Unification and Extrapolation of lumping species

To predict the SOA formation from linear alkanes larger than C12, for which MCM mechanisms do not exist, the product

distributions of C9-C12 are mathematically parameterized using an incremental volatility coefficient (IVC) as a function of

the carbon length and then extrapolated. As the carbon number of linear alkanes increase, the vapor pressure of both the

precursor HC and the resulting oxidation products will decrease, causing a shift in the product distribution. The IVC is meant

to generalize this shift by describing the drop in vapor pressure caused by the addition of a carbon atom. The IVC ($I(n)$)

which is used to calculate the product distribution for a linear alkane with carbon number $n$ using the product distribution of

a linear alkane with carbon number $n$-$1$ is as follows:

$$I(n) = a \left(\frac{1}{b}\right)^{(n-9)} \tag{4}$$

where $a$ and $b$ are obtained by fitting the predicted $\alpha_i$ values using equation 4 to those constructed from the individual HC by

using explicit gas mechanisms in section 3.1. In this study, $a$ and $b$ were calculated to be 1.15 and 1.2, respectively. The $\alpha$

value ($\alpha_{n,j,k}$) with carbon number $n$, at a given volatility (j) and reactivity (k) is predicted by using $\alpha$ value with carbon

number $n$-1 as follows:

$$\alpha_{n,j,k} = I(n) * \left(\alpha_{n-1,j+1,k}\right) + \left(1 - I(n)\right) * \left(\alpha_{n-1,j,k}\right) \tag{5}$$

Term $\alpha_{n-1,j+1,k}$ represents the stoichiometric coefficient of volatility group $j+1$ adjacent volatility group $j$ at a given

reactivity k with carbon number n-1. Term $\alpha_{n-1,j,k}$ represents the stoichiometric coefficient of volatility group $j$ at a given

reactivity k with carbon number $n$-$1$.  Volatility group $j$ is less volatile than volatility group $j+1$. Notably, the shifts in

composition are only within the vapor pressure groups ($j$) at a given reactivity group ($k$).

Using this IVC, the product distribution of C9 is the base (Eq. 4) to predict the product distribution of C10. This process is

sequentially repeated for higher carbon numbers. Each repetition will yield the product distribution of a linear alkane with 1

carbon number larger than the last. Fig. S2 illustrates the construction of product distribution based on the incremental

volatility approach.





### 3.4 SOA formation by partitioning

Partitioning coefficients for each lumping species, $i$, between $g$ and $or$ phase ($K_{or,i}$) or between the $g$ and wet $in$ phase ($K_{in,i}$) are calculated using the typical gas-particle partitioning model (Pankow, 1994):

$$K_{or,i} = \frac{7.501RT}{10^9 MW_{or}\, \gamma_{or,i} p_{L,i}^o} \tag{6}$$

$$K_{in,i} = \frac{7.501RT}{10^9 MW_{in}\gamma_{in,i} p_{L,i}^o} \tag{7}$$

where R is the gas constant (8.314 J mol$^{-1}$ K$^{-1}$), and T is temperature (K). MW$_{or}$ and MW$_{in}$ are the average molecular weights (g mol$^{-1}$) of the organic and inorganic phases of the aerosol, respectively. $p_{L,i}^o$ is the subcooled liquid vapor pressure of a species, $i$. The activity coefficient in the organic phase for each lumping species, $\gamma_{or,i}$, is assumed to be unity (Jang and Kamens, 1998). The activity coefficient in the inorganic phase for each lumping species, $\gamma_{in,i}$, is predicted by a semi-

empirical regression equation which was fit to the activity coefficients of various organic compounds as a function of physicochemical parameters (MW, O:C ratio, and HB) and sulfate fraction (FS). FS is an indicator for aerosol acidity which is defined as follows:

$$FS = \frac{[SO_4^{2-}]}{[SO_4^{2-}] + [NH_4^+]} \tag{8}$$

where $[SO_4^{2-}]$ and $[NH_4^+]$ are the concentration of sulfate and ammonium ions, respectively. The semi-empirical equation,

derived from activity coefficients estimated using the Aerosol Inorganic-Organic Mixtures Functional Groups Activity Coefficients (AIOMFAC) model (Zuend et al., 2011) at a given RH is as follows:

$$\gamma_{in,i} = e^{0.035 \cdot MW_i - 2.704 \cdot \ln(O:C_i) - 1.121 \cdot HB_i - 0.330 \cdot FS - 0.022 \cdot (\cdot RH)} \tag{9}$$

Further information on the derivation and statistical properties of Eq. (9) can be found in Zhou et al. (2019). The partitioning coefficients are used to calculate the concentration of each lumping species in the three phases ($C_{g,i}$, $C_{or,i}$, and $C_{in,i}$) from the

total concentration of each lumping species ($C_{T,i}$). The total SOA mass formed by partitioning in both $or$ and $in$ phases ($OM_P$) is predicted by the following equation which was developed by Schell et al. (2001) and reconstructed to consider mass formed by particle-phase reactions (OM$_{AR}$) by Cao and Jang (2010):

$$OM_P = \sum_i \left[ C_{T,i} - OM_{AR,i} - C_{g,i}^* \frac{\frac{C_{or,i}}{MW_i}}{\sum_i \left(\frac{C_{or,i}}{MW_i} + \frac{OM_{AR,i}}{MW_{oli,i}}\right) + \frac{OM_0}{MW_{oli,i}}} \right] \tag{10}$$




where Cg* ($1/K_{or,i}$) and $OM_0$ (mol m$^{-3}$) represent the effective saturation concentration and preexisting OM, respectively.

$MW_{oli,i}$ and $MW_i$ represent the molecular weights of oligomeric products and lumping species, respectively. Eq. (10) is solved

using Newton-Rapson method, which iterates until a convergence is reached (Press et al., 1992):

### 3.5 SOA formation by particle-phase reactions

The SOA formation from particle-phase reactions (OM$_{AR}$) is attributed to both the *or* and *in* phases. In the *or* phase, SOA

formation is attributed to oligomerization as organic compounds undergo self-dimerization reactions (Odian, 2004). In the *in*

phase, oligomerization of organic compounds can be acid-catalyzed (Jang et al., 2002). Oligomerization reactions are

expressed as second-order reactions with rate constants $k_{AR,or,i}$ and $k_{AR,in,i}$ (L mol$^{-1}$ s$^{-1}$) in the *or* and *in* phases, respectively.

These oligomerization rate constants are used to predict a change in aerosol mass in each phase due to particle phase

reactions over time (Zhou et al., 2019; Im et al., 2014). A semi-empirical model developed by Jang et al. (2005) is used to

estimate $k_{AR,in,i}$  as follows:

$$k_{AR,in,i} = 10^{0.25pK_{BH_i^+}+1.0X+0.95R_i+\log(a_w[H^+])-2.58} \tag{11}$$

where R$_i$ represents species reactivity, $pK_{BH^+_i}$ represents the protonation equilibrium constant, $a_w$ represents the activity of

water, $X$ represents excess acidity (Cox and Yates, 1979), and $[H^+]$ represents the concentration of protons which are

estimated using the extended aerosol inorganic model (E-AIM (Clegg et al., 1998)) $k_{AR,or,i}$ is determined as follows:

$$k_{AR,or,i} = 10^{\left[0.25pK_{BH_i^+}+0.95R_i+1.2\left(1-\frac{1}{1+e^{0.005(300-MW_{or})}}\right)+\frac{2.2}{1+e^{6(0.75-O:C)}}-10.07\right]} \tag{12}$$

For the oligomerization in the *or* phase, the terms related to acidity ($X$, and $a_w[H^+]$) are excluded. Studies have previously

demonstrated that aerosol viscosity can influence the mobility of chemical species and thus, apparent reaction rates, which

can be limited by slow bulk diffusion in the particle-phase (De Schrijver and Smets, 1966; Reid et al., 2018). The molecular

weight of species in the organic phase (MW$_{or}$) and the O:C ratio, which are important predictors for viscosity, are considered

to calculate $k_{AR,or,i}$ (Han and Jang, 2022).



### 3.6 Organosulfate formation

In the wet *in* phase of the aerosol, sulfuric acid can react with reactive organic compounds to form dialkyl sulfate (diOS). This formation of diOS can contribute to SOA mass production but can also lead to a reduction in [H⁺] which decrease the

250 rate of SOA mass produced by acid-catalyzed oligomerization in the *in* phase. The UNIPAR model predicts diOS formation in relation to the concentration of free sulfate ($[SO_4^{2-}]_{free}$) that is unassociated with ammonium.

$$[SO_4^{2-}]_{free} = [SO_4^{2-}] - 0.5[NH_4^+] \tag{13}$$

This free sulfate concentration can then be used in a semi-empirical equation developed by Im et al. (2014) to calculate the fraction of free sulfate that will form diOS as follows:

$$\frac{[diOS]}{[SO_4^{2-}]_{free}} = 1 - \frac{1}{1 + f_{diOS} \frac{N_{diOS}}{[SO_4^{2-}]_{free}}} \tag{14}$$

where $f_{diOS}$ is a semi-empirical conversion factor that was introduced by Im et al. (2014). In this study $f_{diOS}$ is 0.04. $N_{diOS}$ is a parameter that represents the sum of functional groups, across all lumped species, that can react to form diOS, scaled by a weighing factor based on ability to react with sulfate.

### 3.7 Correction of intermediate organic vapor deposition to walls

In addition to wall loss of precursor HC, intermediate oxidized products derived from the precursor HC can deposit to the wall. The organic vapor deposition to wall is kinetically described with the deposition ($k_{on,i}$) and desorption ($k_{off,i}$) rate constants of each lumping species, *i*, to the UF-APHOR wall by using the method detailed by Han et al. (2019). $k_{on,i}$ is expressed as a fractional loss rate (Mcmurry and Grosjean, 1985):

$$k_{on,i} = \left(\frac{A}{V}\right) \frac{\alpha_{w,i}\bar{v}_i/4}{1 + \frac{\pi \alpha_{w,i}\bar{v}_i}{8(K_e D)^{1/2}}} \tag{15}$$

where $D$ ($1.0 \times 10^{-6}$ m² s⁻¹) and $K_e$ (0.12 s⁻¹) are the diffusion coefficient and coefficient of eddy diffusion applied as a fixed number, respectively. $\left(\frac{A}{V}\right)$ represents the surface area to volume ratio of the chamber. $\bar{v}_i$ and $\alpha_{w,i}$ represent the mean thermal speed of the gas molecules, and accommodation coefficient of *i* to the wall, respectively. Equations for the calculation of $\bar{v}_i$ and $\alpha_{w,i}$ can be found in section S4. $K_{w,i}$ ($K_{w,i} = k_{on,i}/k_{off,i}$) is calculated as follows:

$$\ln(K_{w,i}) = -\ln(\gamma_{w,i}) - \ln(p_{L,i}^o) + \ln\left(\frac{7.501 RT OM_{wall}}{10^9 MW_{OM}}\right) \tag{16}$$



where $p_{L,i}^o$ (mmHg) is the liquid vapor pressure of each lumping group, $i$. $R$ (8.314 J mol$^{-1}$ K$^{-1}$) is the ideal gas constant and

T (K) is temperature. $OM_{wall}$ (mg m$^{-3}$) and $MW_{OM}$ are the concentration of organic matter on the wall, and the molecular

weight of organic matter on the wall, respectively. The activity coefficient, $\gamma_{w,i}$, is calculated using the quantitative structure-

activity relationship (QSAR) approach with the physicochemical properties $H_{d,i}$, $H_{a,i}$, and $\alpha_i$ which represent hydrogen bond

acidity, hydrogen bond basicity, and polarizability of each lumping group $i$, respectively (Abraham et al., 1991; Abraham

and Mcgowan, 1987; Leahy et al., 1992; Platts et al., 1999; Puzyn et al., 2010). Eq. (16) can be rewritten as:

$$\ln(K_{w,i}) = -(a_p H_{d,i} + b_p H_{a,i} + r_p \alpha_i + c_p) - \ln(p_{L,i}^o) + \ln\left(\frac{7.501 RT OM_{wall}}{10^9 MW_{OM}}\right) \tag{17}$$

The values of $H_{d,i}$, $H_{a,i}$, and $\alpha_i$ (Table S2) were calculated using the PaDEL-Descriptor, (Yap, 2011). The value of $K_{w,i}$ is used

along with the $k_{on,i}$ to predict intermediate product using an analytical equation from the study by Han et al. (2019) wall loss

as follows:

$$C_{g,i} = \frac{K_{w,i} C_{T,i}}{K_{w,i}+1} e^{-k_{on,i}\left(1+\frac{1}{K_{w,i}}\right)t} + \frac{C_{T,i}}{K_{w,i}+1} \tag{18}$$

where $C_{g,i}$ (µg m$^{-3}$) is the gas-phase concentration of a lumping species, $i$, after time step $t$ (360 s). $C_{T,i}$ (µg m$^{-3}$) is the sum of

$C_{g,i}$ and the concentration of lumping species $i$ on the chamber wall ($C_{w,i}$ (µg m$^{-3}$)).

### 3.8 UNIPAR procedure for SOA mass production each time step

At each step, $C_{T,i}$ is estimated by using the newly introduced ΔHC and $\alpha_i$ and is combined with the previous step's

concentration of lumping species except those used for the formation of OM$_{AR}$ and organic vapor deposition to walls for the

simulation of chamber data. Then, the updated $C_{T,i}$ is split into $C_{g,i}$, $C_{or,i}$, and $C_{in,i}$ based on multiphase partitioning

coefficients as seen in Eq. (6) and Eq. (7). $C_{or}$ and $C_{in}$ are then used to form $OM_{AR}$ via oligomerization in both the *in* and *or*

phases with the rate constants calculated in Eq. (11) and Eq. (12), respectively. In the model, the quantity of the sulfate

associated with the esterification of sulfuric acid to form organosulfates in the *in* phase is also estimated (Eq. 14). Following

the process to form $OM_{AR}$, the remaining concentration of lumping species is used to estimate the organic vapor deposition to

the wall using Eq. (18). OM$_P$ is calculated using a Newtonian approach (Eq. 10) in the presence of OM$_{AR}$ and the preexisting

OM$_0$ at the end of each time step. For the total SOA mass, OM$_{AR}$, OM$_P$ and OM$_0$ are combined.



## 4 Results and Discussion

### 4.1 Chamber data vs. model prediction

UNIPAR model predictions were compared to data collected under various environmental conditions in the UF-APHOR

chamber. As seen in Fig. 2, the C10 SOA mass prediction based on the product distributions created using the preexisting

MCM was almost negligible and significantly underestimated from observation. The addition of the alkyl peroxy radical

autoxidation mechanism (Section 3.1) was able to considerably improve the prediction of alkane SOA formation. Fig. 2

obviously indicates that alkane SOA mass is mainly attributed to low volatility products produced from alkyl peroxy-radical

autoxidation.

Fig. 3 together with Fig. 2 shows that UNIPAR, equipped with product distributions derived from the MCM mechanisms

with alkyl peroxy radical autoxidation reactions, can reasonably predict final SOA mass produced under various

environmental conditions for alkanes C9, C10, and C12. For C11, the model is able to well predict the low $NO_x$ condition

with sulfuric acid seed (C11C) but shows an over prediction for the other conditions (C11A and C11B). Additionally, the

model is able to well predict SOA mass in the presence of acidic seed as seen in C9C, C9D, C10B, C10D, and C11C. The

detailed discussion on the impact of acidic seed is found in section 4.3 for model sensitivity.

### 4.2 SOA prediction via product distributions with IVC

To predict the SOA formation from a wide range of alkane lengths, the product distributions of linear alkanes were

mathematically predicted as a function of carbon number using an IVC as discussed in section 3.3.  The resulting product

distributions were applied to the UNIPAR model (Fig. 4).  Compared to the prediction form the product distribution with

MCM and alkyl peroxy radical autoxidation (Fig. 3), the unified simulation of C10 and C11 in Fig. 4 was significantly

improved. C12B and C12C were somewhat improved in the simulation with the unified stoichiometric coefficient array but

C12A was overpredicted. It is possible that the cold temperature for experiment C12A (Table 1) caused significant organic

vapor wall loss that was unaccounted for in this simulation.

### 4.3 Extrapolation of UNIPAR to longer chain alkanes

The simulation of alkane SOA using the IVC-based product distributions, which were demonstrated with C10-C12 alkanes in section 4.2, was evaluated for prediction of C13 and C15 SOA. Because the explicit mechanisms for the alkanes higher than C12 are not currently available in MCM, the CB6 ozone model (Emery et al., 2015) was implemented to predict the consumption of alkane and the concentrations of $HO_2$ and $RO_2$ and used with the UNIPAR model integrated with the IVC-based product distributions. In Figure 5, the simulated C13 and C15 SOA masses were compared to chamber data. Overall, SOA mass was reasonably predicted by using the UNIPAR model integrated with the IVC-based product distribution. Furthermore, SOA simulation was improved when considering the organic-product vapor deposition to the chamber wall. The impact of the chamber wall artifact is generally greater with the longer chain alkanes. For example, the gap between the simulations of SOA mass with and without the wall artifact is larger for C15 compared to C13 as seen in Fig. 6. Notably, the UNIPAR model is unable to predict the early peak in SOA formation that is seen in the chamber data for C12 (Fig. 5), C13 and C15 alkanes (Fig. 6). This initial spike in observed SOA mass is likely because of the fast SOA formation due to the reaction with a high concentration of hydroxyl radicals, which is produced via photolysis of the HONO introduced into chamber. This initial spike is generally larger with the longer alkane. The low and mid-volatility products that are formed in the morning will be rapidly condensed via nucleation and non-equilibrium partitioning. The viscosity of the oxidized products tends to be very high, especially for larger alkanes, due to the high molecular weight and the low O:C ratios. The redistribution of organic products between the gas and particle phase to reach equilibrium is achieved via their slow evaporation. This will typically not occur in the ambient atmosphere as the hydroxyl radical is gradually produced via the small amount of HONO in the morning or the ozone mechanism in daytime.

### 4.4 Sensitivity of SOA formation to $NO_x$ levels, temperature, $OM_0$, and seed condition

Fig. 6 illustrates the sensitivity of 6 different alkane SOA yields to 6 $HC/NO_x$ levels at a given temperature (298K) under the same sunlight profile (Fig. S3) obtained on 01/20/20 and 30% RH. All sensitivity simulations for a series of alkanes (C10, C12, C14, C16, C18, and C20) were performed with the CB6-UNIPAR model (Table S3) integrated with the IVC-base unified product distribution. The impact of $NO_x$ conditions on linear alkane SOA formation has not been studied extensively

in the past. In general, aromatic SOA yields rapidly increase with decreasing the NOx level up to HC ppbC/NOx ppb ≈ 5,

reaching to a plateau (Im et al., 2014), while SOA yields for each alkane generally increased as the HC /$NO_x$ level increased

over a broad range of the HC ppbC/NOx ppb ratio. Zhang and Seinfeld (2013) found that SOA yields for Dodecane (C12)

were higher in lower $NO_x$ conditions.  Cappa et al. (2013) found significant differences in SOA yields between high and low

$NO_x$ conditions after 10 hours of oxidation, with higher yields for the low $NO_x$ condition. Notably, Zhang and Seinfeld

(2013) and Cappa et al. (2013) use $H_2O_2$ as a source of hydroxyl radicals for the low $NO_x$ condition, while the chamber

experiments in our study generate hydroxyl radicals using HONO for both low and high $NO_x$ conditions.

As discussed in Section 4.1, a significant part of alkane SOA is comprised of the low volatility products originating from

autoxidation mechanisms. The fraction of autoxidation products to the total SOA mass generally decreases with increasing

carbon number or increasing the $NO_x$ level due to gas-particle partitioning of non-autoxidation lowly volatile products as

seen in Table S4.

Fig. 7 shows the simulated SOA yields, using UNIPAR with IVC-based product distributions, for a series of alkanes at 3

different temperatures (278K, 288K, 298K) and 2 different $NO_x$ levels (HC ppbC/$NO_x$ ppb= 15, 3) at $OM_0$ = 5.  For both

high and low $NO_x$ levels, each alkane produces a higher SOA yield at a lower temperature. The volatility of non-autoxidation

products from the larger alkane such as eicosane (C20) is low enough to exist primarily in the particle phase, causing the

lower sensitivity to temperature compared to smaller alkanes. This result agrees with other studies that have shown a

significant impact of temperature on the SOA formation of n-dodecane and n-undecane under low $NO_x$ conditions (Li et al.,

2020; Takekawa et al., 2003). For H$NO_x$ conditions, the result also agrees with past work that has found a decent impact of

temperature on the SOA formation from n-dodecane at the low $OM_0$, which is relevant to ambient aerosol (Lamkaddam et

al., 2017). Fig. 7 also includes the contribution of $OM_P$ and $OM_{AR}$ to SOA in the absence of inorganic seed.  Overall, the

contribution of $OM_{AR}$ is higher with lower temperatures and higher carbon numbers.

Figs. S4 and S5 illustrate the sensitivity of alkane SOA to various preexisting aerosol mass ($OM_0$) ranging from 2.5 to 10

µg/m$^3$ and aerosol acidity (no seed, wet-AS, and wet-AHS at 60%RH), respectively. SOA yields are significantly increased

for each linear alkane as the initial organic matter in the simulation increases. Overall, smaller alkanes have a greater

sensitivity to $OM_0$.  Alkane oxidation products are generally hydrophobic with a low O:C ratio and very weakly soluble in



the wet *in* phase. For example, the simulated O:C ratios of alkane SOA in this study ranges from 0.48 and 0.55. Thus, alkane SOA is insignificantly impacted by reactions in the wet *in* phase, as seen in Fig. S4, but significantly influenced by gas-aerosol partitioning.

### 4.5 Uncertainty of model parameters

In the autoxidation mechanism for C12, three products originating from autoxidation of an alkyl peroxy radical (SR1) dominate the formation of low volatility products to form SOA mass as seen in Table S5: i.e., SR3, SR34, and SR40. In order to the evaluate the uncertainty in SOA formation via autoxidation, the rate constant for the 1st step of autoxidation of 370 an alkyl peroxy radical (SR1) was increased/decreased by 50%. Fig. 5 illustrates the resulting mass changes due to change in this rate constant(SR1): 7.5% increase and 15.7% decrease in SOA mass formed by increasing/decreasing by 50%, respectively for 100 µg/m$^3$ C12 consumption at HC ppbC/NO$_x$ ppb = 3, and RH = 30%. The rate constant ($k_{AR,in,i}$) of acid-catalyzed oligomerization of organic species $i$ in inorganic phase was increased/decreased by a factor of 2 but the resulting SOA mass shows no significant change for 100 µg/m$^3$ of C10, C12, C14, C16, C18, and C20 consumptions under the high 375 NO$_x$ conditions (HC ppbC/NO$_x$ ppb = 3). This is expected as alkane oxidation products are lowly soluble in salted *in* phase with low reactivity.

### 4.6 Atmospheric Implication

Several studies in the regional scale simulation have considered alkanes as an important source (Hodzic et al., 2010; Lee-380 Taylor et al., 2011; Pye and Pouliot, 2012; Yang et al., 2019). In this study, the UNIPAR model was employed to predict SOA formation from linear alkanes originating from both gas-particle partitioning and particle-phase reactions. Unlike the SOA formation from other precursors (aromatics and biogenics) in which OM$_P$ and OM$_{AR}$ both are significant, alkane SOA is predominantly produced via gas-particle partitioning of organic products. The inclusion of alkyl peroxy-radical autoxidation reactions in the gas mechanism was found to significantly improve SOA mass predictions as seen in Fig. 3. 385 Furthermore, the product distributions created using the MCM mechanisms with alkyl peroxy radical autoxidation reactions



were extended to larger alkanes using the IVC approach (Fig. S2). Then, these product distributions were coupled with organic vapor deposition to walls to predict chamber data for C13 and C15 (Fig. 6).

$NO_x$ levels and temperature were found to be the two most influential factors on alkane SOA production (Fig. 6 and 7). At lower $NO_x$ levels, the SOA yield from each of C10, C12, C14, C16, C18, and C20 were found to be higher compared to higher $NO_x$ levels. The impact of temperature was found to be more significant for smaller alkane species as they were more likely to have intermediate volatility products which have partitioning processes that are sensitive to changes in temperature. No significant seed impact appeared regardless of any acidity (Fig. S5). Thus, the reduction of $SO_2$ emissions would not have a significant impact on linear alkane SOA formation. Furthermore, the reduction of $NO_x$ emissions would serve to increase the alkane SOA yields in ambient air. Although the impact of aerosol-phase reactions on the SOA mass created by linear alkane precursors is insignificant, the SOA mass produced by aerosol-phase reactions from products of other precursors (aromatics and biogenics) in the ambient environment can serve to increase partitioning of the products derived from alkanes. Accordingly, each of the series of linear alkanes, ranging from C10-C20, were sensitive to concentrations of pre-existing $OM_0$ (Fig. S4).

Diesel fuel is comprised of various lengths of linear alkanes ranging from C9 to C24. In order to simulate diesel SOA formation, the composition of mass weighted linear alkanes (Gentner et al., 2012) in diesel fuel was applied to UNIPAR model in Fig. 8. About 80% SOA mass from diesel linear alkanes is produced from long-chain alkanes (≥C15), with the highest percentage (≈11.8%) from C18, while long chain gaseous alkanes comprise only about 60% of the linear alkane mass. Thus, we conclude that long-chain alkanes are an important source of SOA formation in urban environments. Gentner et al. (2012) also predicted SOA formation from various alkanes in diesel and found that long-chain alkanes make up the vast majority of the SOA mass. Notably, Gentner et al. (2012) found that the peak of SOA mass production from linear alkanes in diesel appears between C19 and C22 which is a slightly higher carbon number compared to our prediction.

In addition to linear alkanes, branched and cyclic alkanes are important components of diesel fuel (Gentner et al., 2012) and must be considered when predicting SOA formation in urban environments. Improvements can also be made on oxidized product distributions originating from linear alkanes. The MCM mechanisms used for linear alkanes in this study are simplified in such a way that, if a specific reaction (i.e. reaction with hydroxyl radical) can occur on multiple locations on a





specific compound, only one instance of this reaction pathway is included. The inclusion of each of these different reaction pathways can possibly modulate the product distribution and ultimately increase/decrease the predicted SOA production from the UNIPAR model.

**Code availability**

Code to run the SOA model in this study is available upon request.

**Data availability**

The chamber data and simulated results used in this study are available upon request

**Author contributions**

MJ designed the experiments, and AM, DD, and MJ carried them out. AM prepared the manuscript with contributions from MJ.

**Competing interests**

The authors declare that they have no conflict of interest.

**Financial Support**

This research was supported by the National Institute of Environmental Research (NIER2021); the National Science Foundation (AGS1923651); and the Fine Particle Research Initiative in East Asia Considering National Differences (FRIEND) Project through the National Research Foundation of Korea (NRF) funded by the Ministry of Science and ICT (2020M3G1A1114556)

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



**Table 1. Summary of experimental conditions and observed data for experiments performed in the UF-APHOR outdoor chamber.**

| ID | Date | Alkanes | Seed type[a] | Seed mass (µg/m³) | Initial HC (ppb) | HC/NOx (ppbC/ppb) | Temp (K) | RH (%) | OM (µg/m³) | Yield[b] (%) | Comments |
|---|---|---|---|---|---|---|---|---|---|---|---|
| C9A | 05/07/21 | C9 | None | 0 | 130 | 7.2 | 286-312 | 34-94 | 2.0 | 0.8 | Figs 3, 4 |
| C9B | 05/07/21 | C9 | None | 0 | 147 | 1.9 | 287-310 | 43-99 | 4.1 | 2.4 | Figs 3, 4 |
| C9C | 05/18/21 | C9 | SA | 81 | 121 | 18.8 | 291-316 | 22-77 | 13.0 | 4.1 | Figs 3, 4 |
| C9D | 05/18/21 | C9 | SA | 100 | 149 | 7.2 | 291-315 | 29-84 | 16.6 | 3.2 | Figs 3, 4 |
| C10A | 05/29/21 | C10 | None | 0 | 136 | 6.4 | 292-317 | 24-81 | 25.6 | 5.2 | Figs 2, 4 |
| C10B | 05/29/21 | C10 | SA | 34 | 140 | 6.9 | 292-317 | 31-90 | 31.6 | 4.3 | Figs 3, 4 |
| C10C | 06/01/21 | C10 | None | 0 | 136 | 2.0 | 292-314 | 17-45 | 37.7 | 5.7 | Figs 2, 4 |
| C10D | 06/01/21 | C10 | SA | 35 | 136 | 1.8 | 292-314 | 20-38 | 28.1 | 4.0 | Figs 3, 4 |
| C11A | 09/05/20 | C11 | None | 0 | 140 | 9.3 | 297-321 | 21-52 | 47.4 | 7.0 | Figs 3, 4 |
| C11B | 09/18/20 | C11 | SA | 52 | 202 | 22.4 | 297-325 | 12-42 | 44.3 | 7.0 | Figs 3, 4 |
| C11C | 09/18/20 | C11 | SA | 60 | 209 | 4.0 | 298-321 | 21-52 | 30.1 | 3.1 | Figs 3, 4 |
| C12A | 01/08/20 | C12 | None | 0 | 374 | 5.4 | 277-302 | 22-72 | 44.1 | 2.8 | Figs 3, 4 |
| C12B | 06/07/22 | C12 | None | 0 | 203 | 2.3 | 293-321 | 23-91 | 19.0 | 2.4 | Figs 3, 4 |
| C12C | 06/07/22 | C12 | None | 0 | 172 | 4.9 | 293-320 | 32-99 | 123.2 | 20.6 | Figs 3, 4 |
| C13A | 10/18/20 | C13 | None | 0 | 207 | 17.1 | 290-315 | 26-89 | 286.9 | 11.3 | Fig 5 |
| C13B | 10/18/20 | C13 | SA | 35 | 202 | 16.6 | 291-315 | 31-95 | 195.8 | 8.7 | Fig 5 |
| C15A | 02/17/22 | C15 | None | 0 | 202 | 4.3 | 288-307 | 30-78 | 388.8 | 46.8 | Fig 5 |
| C15B | 02/17/22 | C15 | None | 0 | 202 | 8.7 | 289-316 | 27-95 | 206.7 | 28.5 | Fig 5 |

[a] Experiments were performed with no seed (none) and sulfuric acid seed (SA). [b] Yield was calculated as the ratio between the
concentration of the final measured SOA mass (µg/m³) and the concentration of precursor alkane consumed (µg/m³).



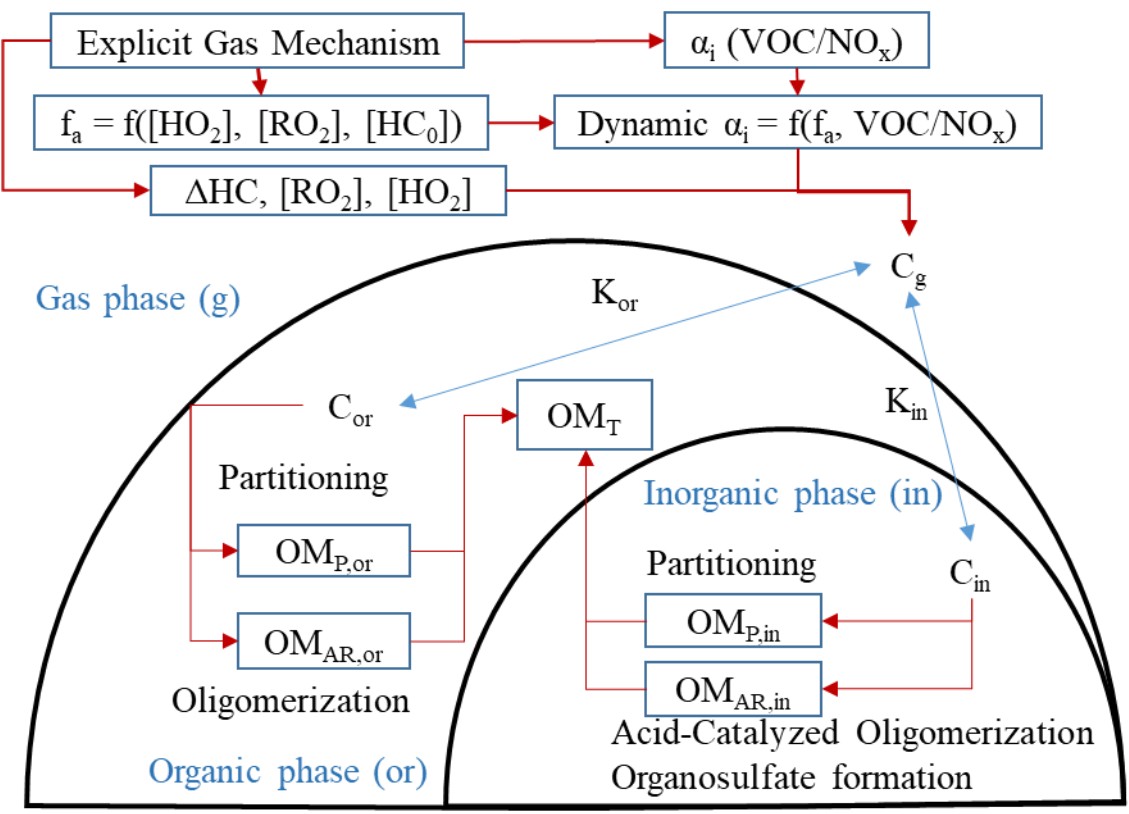

**Figure 1. A simplified scheme of the UNIPAR model. $[HC]_0$ represents the initial hydrocarbon (HC) concentration. A gas kinetic mechanism (MCM v3.3.1 + autoxidation of alkyl peroxy radicals) is used to simulate the consumption of HC ($\Delta HC$) and the concentrations of hydroperoxide radical ($[HO_2]$) and organic peroxyl radical ($[RO_2]$). The explicit gas mechanism of each alkane is used to derive the dynamic mass-based stoichiometric coefficient (dynamic $\alpha_i$) of lumping species $i$ as a function of $HC/NO_x$ and the aging factor $f_a$. $f_a$ is represented as a function of $[HO_2]$, $[RO_2]$, and $[HC]_0$ (Zhou et al., 2019). Subscripts $g$, $or$, and $in$ denote the gas, organic, and inorganic phases, respectively. $K_{or}$ and $K_{in}$ represent the partitioning coefficients of lumping species to the $or$ phase and $in$ phase, respectively. $C_{or}$ and $C_{in}$ represent the concentrations of lumping species in the $or$ and $in$ phases, respectively. $OM_{P,or}$ and $OM_{P,in}$ represent the mass of organic matter (OM) present in $or$ and $in$ phases, respectively due to partitioning. $OM_{Ar,or}$ and $OM_{AR,in}$ represent the OM formed in the $or$ phase due to in-particle chemistry such as oligomerization, and $in$ phase due to acid-catalyzed oligomerization and organosulfate formation (Beardsley and Jang, 2016; Im et al., 2014; Zhou et al., 2019). $OM_T$ represents the total SOA mass formed due to partitioning and aerosol-phase reactions in both the $in$ and $or$ phases.**



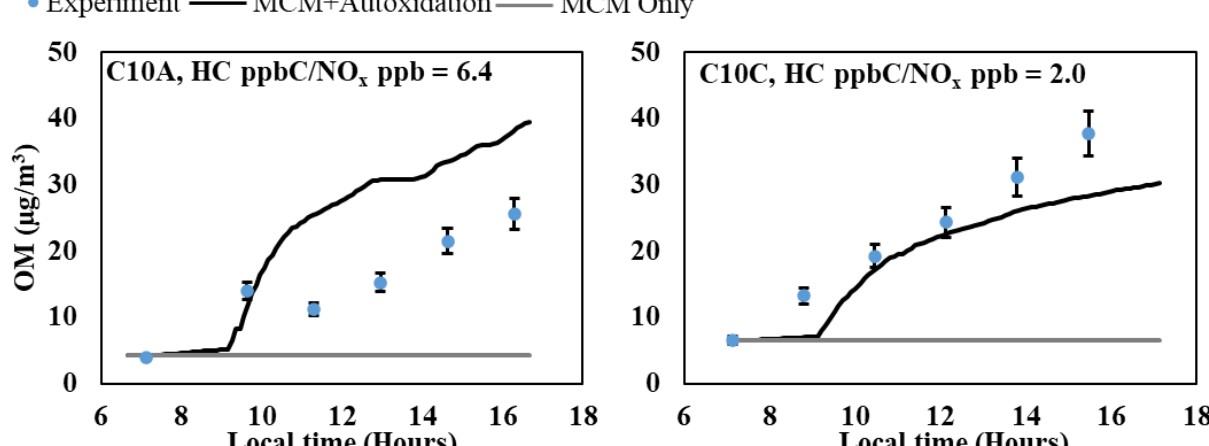

**Figure 2. Comparison of SOA mass produced between simulations using a product distribution for C10 derived from the MCM (gray line) and one from the MCM with alkyl peroxy radical autoxidation reactions added (black line). The blue dots represent observed SOA data collected that are corrected for particle wall loss to the chamber. Error bars**
**represent 9% uncertainty associated with the OCEC and particle wall loss correction.**





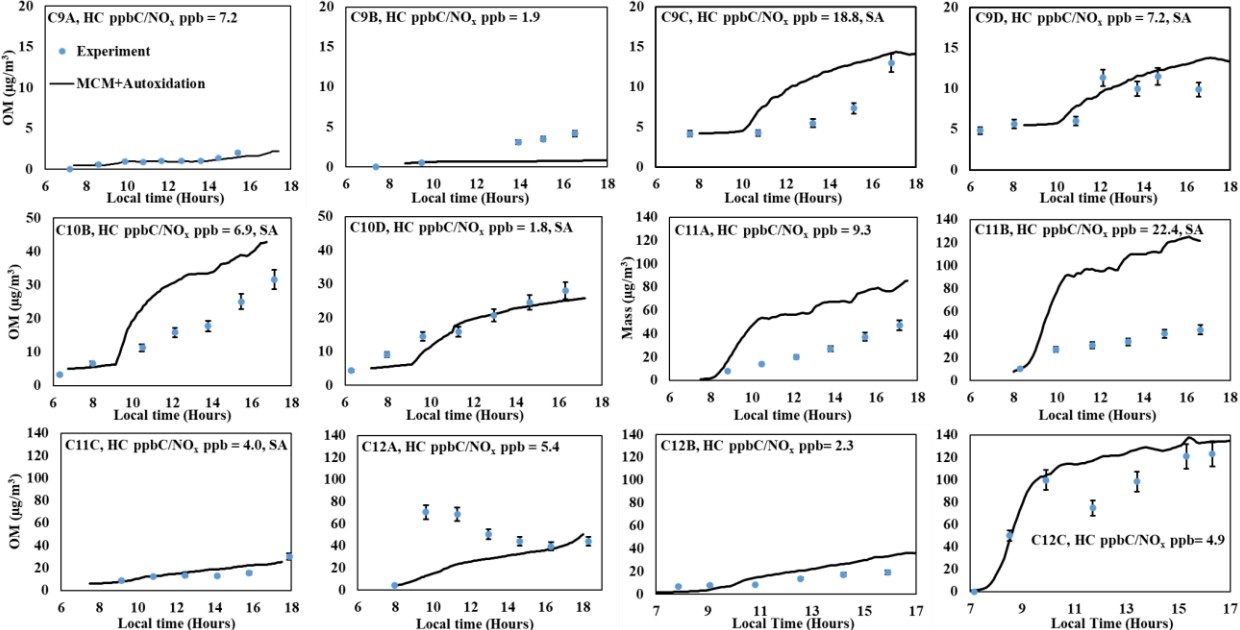

**Figure 3. Simulated SOA mass for alkanes C9, C10, and C12 using a product distribution derived from the MCM mechanisms with alkyl peroxy radical autoxidation reactions added. The blue dots represent observed SOA data collected that are corrected for particle wall loss to the chamber. Error bars represent 9% uncertainty associated with the OCEC measurement and particle wall loss correction.**





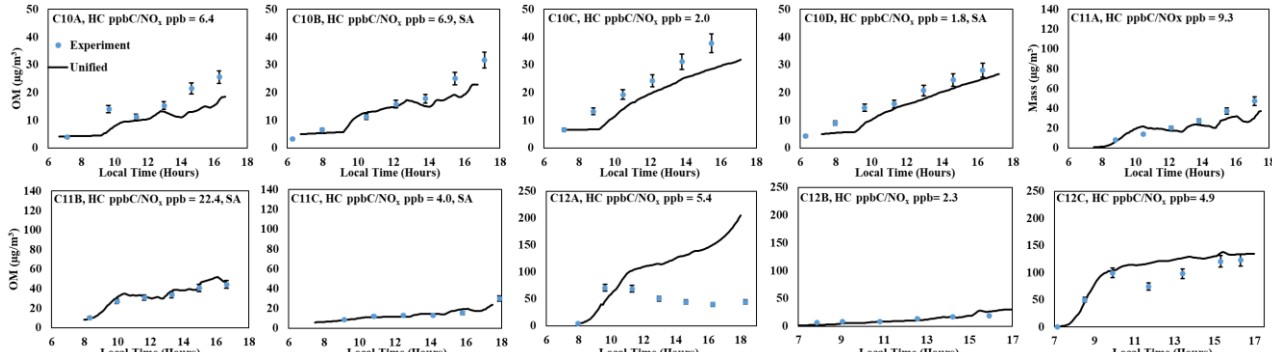

**Figure 4. Comparison of SOA mass produced between simulations, for C10, C11, and C12, using the unified product distributions based on the IVC (black line) to chamber data. The blue dots represent observed SOA data collected that are corrected for particle wall loss to the chamber. Error bars represent 9% uncertainty associated with the OCEC and particle wall loss correction.**





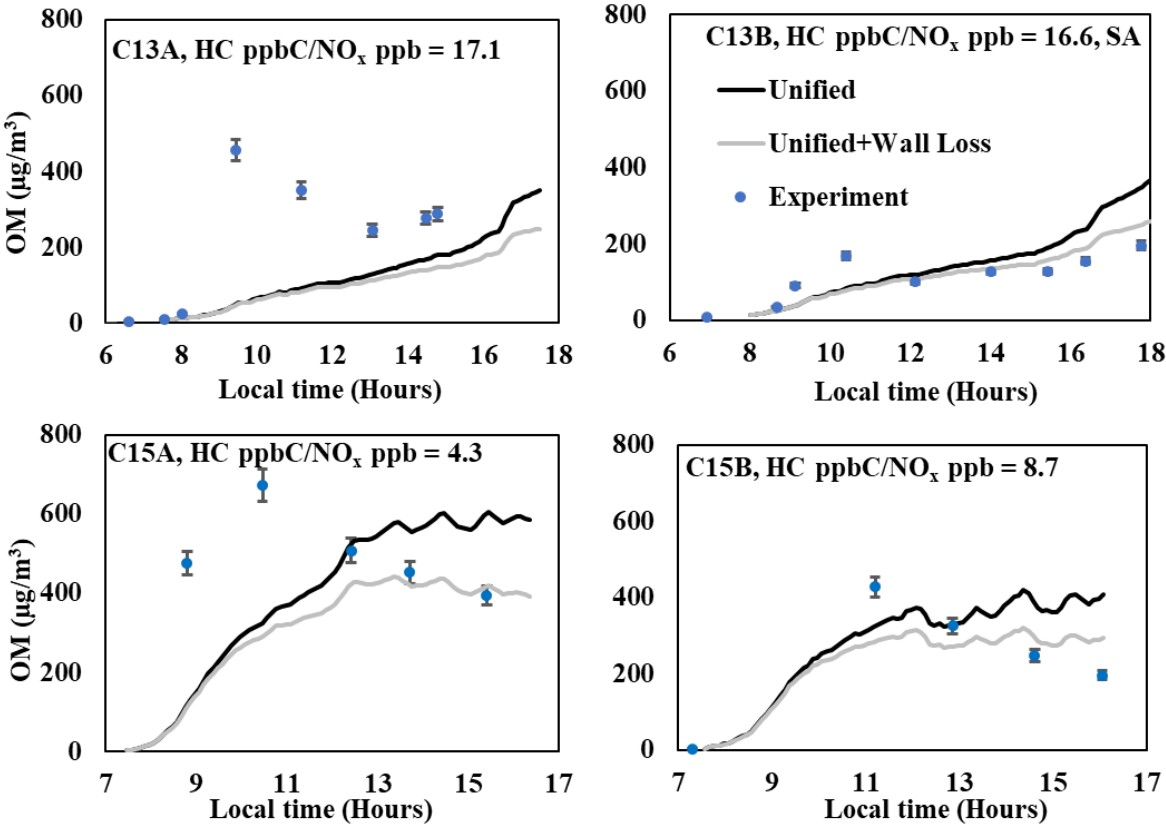

**Figure 5. Comparison of the observed and the predicted SOA mass produced from the photooxidation of C13 and C15 alkanes under different experimental conditions. The SOA simulation was performed by using the unified product distributions that is constructed with the IVC approach in the presence of precursor vapor deposition to the chamber wall. In addition, SOA simulation was performed in the presence (gray line) and the absence (black line) of organic product loss to the wall. The observed SOA mass (dots) was corrected for gas dilution and the particle loss to the chamber wall. ASCM was used to measure SOA mass concentrations for C13 and C15 instead of OCEC data as the artifacts via off-gassing of semi-volatiles to the denuder upstream OC for C13 and C15 oxidation products. The collection efficiency of organic alkane aerosol for the ACSM was assumed to be 66% for C13 and 63% for C15. The collection efficiency was determined by comparison SMPS data with ACSM data with the aerosol originating from non-volatile organics. The collection efficiency of C13B aerosol, which was produced in the presence of acidic seed, was assumed to be 100%. The error associated with OM (6%) is calculated with ACSM data.**




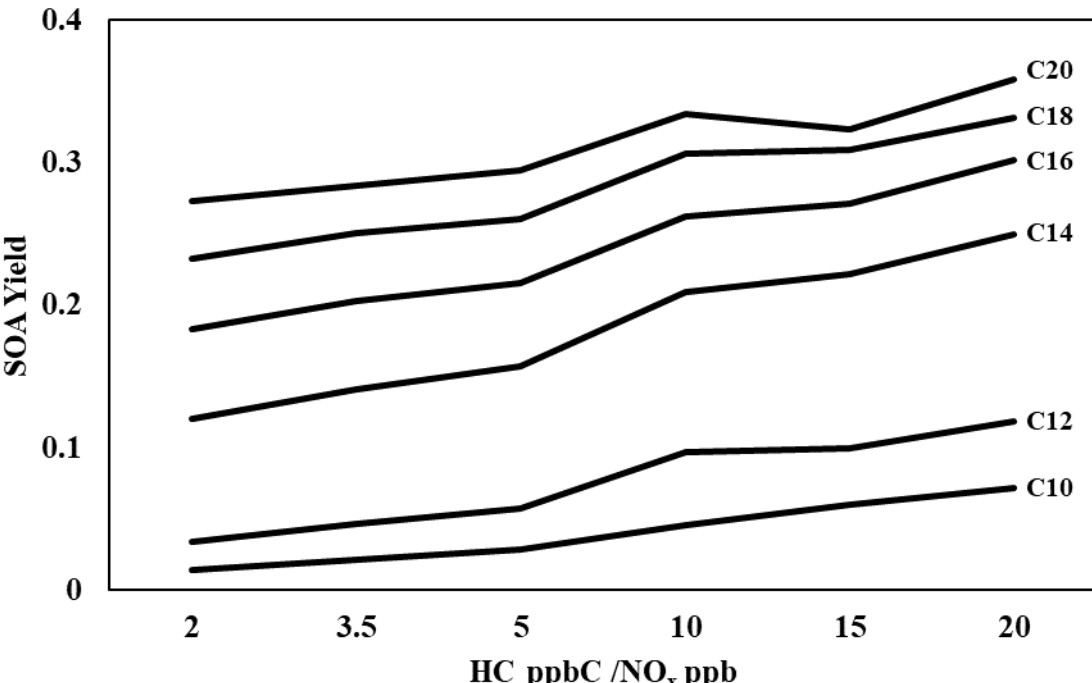

**Figure 6. Simulated SOA Yields for a series of linear alkanes at 6 different HC/NO$_x$ conditions using the IVC-based product distributions. OM$_0$ = 5 µg/m$^3$, 298 K, RH = 30%.**





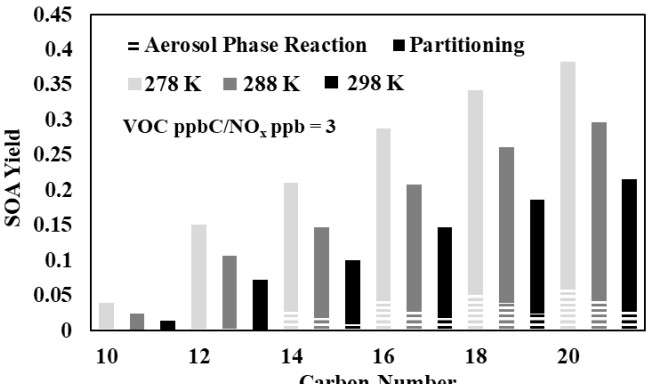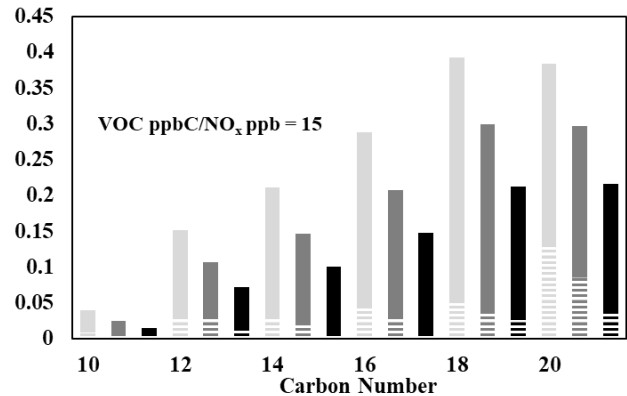

**Figure 7. Simulated SOA Yields, separated by $OM_P$ and $OM_{AR}$, for a series of linear alkanes at 3 different temperatures and 2 different HC ppbC/$NO_x$ ppb conditions using the IVC-based product distributions. $OM_0$ = 5 μg/m³, RH = 30%.**

.




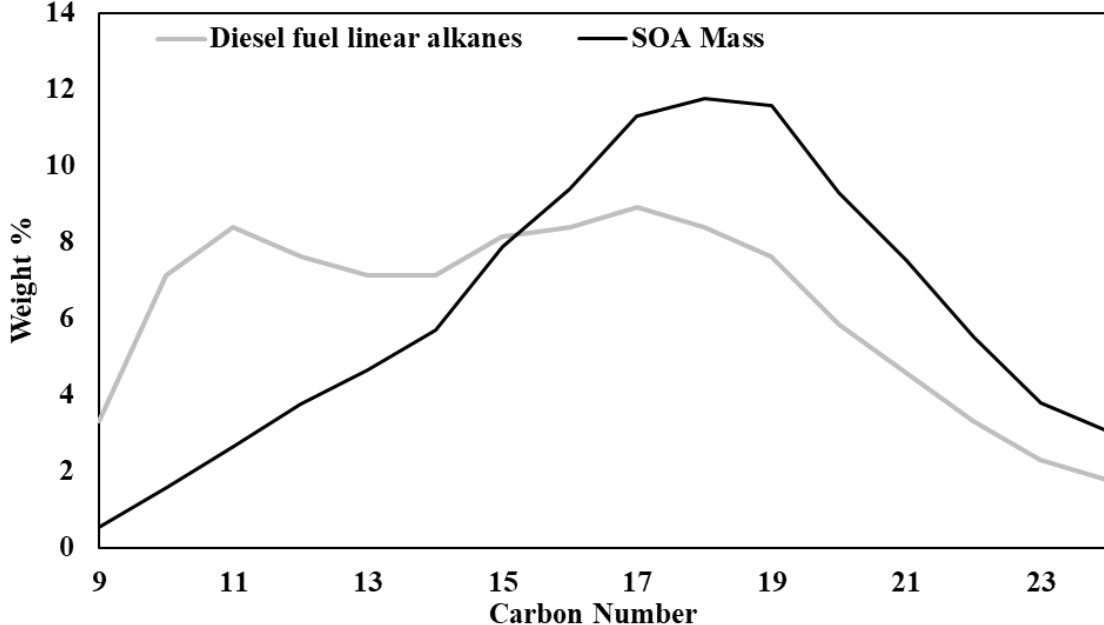

**Figure 8. The predicted relative SOA mass formation from the composition of linear alkanes in diesel fuel. The simulation was performed using the CB6-UNIPAR model equipped with IVC-based product distributions. The relative composition of linear alkanes in diesel fuel used is from Gentner et al. (2012). HC ppbC /NO$_x$ ppb = 3, RH = 60%, OM$_0$ = 5 μg/m$^3$.**
