# Peer review of "Modeling the Influence of Chain Length on SOA Formation via Multiphase Reactions of Alkanes"

_EGUsphere, 2022_

## Author Response (AR1)

**General Notes**

During the revision, it was discovered that the calculation factor used to measure the initial concentration of C10 was wrong. This factor has been corrected and the initial concentrations for C10 have been updated. To improve the model prediction, the coefficient of reaction SR1 (Table S4), and analogous reactions for C9, C10, C11, have been updated in revised manuscript.

**Response to Reviewer 1**

Thank you for reviewing this paper. We hope we have appropriately addressed all of your concerns below.

1. *This work is a development of a previous work by some of the same workers (Zhou et al 2019), and it is not always immediately apparent what is new work and what is a restatement of that previous work. The text should clarify the distinction. The restated sections should be condensed and appropriate reference should be made to the prior work.*

   Where possible, the model description has been condensed. In other instances, appropriate references to previous work has been added early in each section to make this more clear.

2. *This study should add at least one sensitivity test that attempts to model the morning spike seen in the chamber observations, to support the authors' assertion about its origin.*

   Unfortunately, the model is unable to capture this phenomenon and thus, a sensitivity test to model this spike is not possible. A reference to a previous work which also shows the same morning spike for another precursor has been added as further support.

3. *Line 300: discussion of Figure 3. I agree that C9D, C10D and C11C show good model-data comparisons. Comparisons C9C and C10B are not so good, and look more like comparisons C11A and C11B, which are said to be poor. The text should be clearer about this distinction. This comment des not detract from the paper's overall conclusions.*

   These simulations have been updated with the new rate constant as noted previously, which has improved the model predictions for C9C, and C10B.

4. *Section S4, Table S2: It is unfortunate that the accommodation coefficient alpha_w,i, and the polarizability alpha_i have such similar symbols, and that these could be confused with the mass-based stoichiometric coefficient alpha_i introduced in section 3.2. It might be appropriate to change one of these symbols, or at least to explicitly acknowledge the possibility for confusion.*

   This similarity has been explicitly addressed within the text. Line 259: "Note that this $\alpha_i$ used to represent polarizability is different from the $\alpha_i$ used to represent the mass-based stoichiometric coefficient introduced in section 3.2."

5. *Line 131-132, 197, 215 etc: This reviewer finds the abbreviations "or" and "in" confusing in the text because they can be mistaken for ordinary words. Please spell out the words "organic" & "inorganic" unless the abbreviations are used in combination with other symbols? Or, at a minimum, give them bold italic font to differentiate them from normal text.*

This has been addressed by replacing with "or" and "in" with "org" and "inorg", respectively.

6. *Line 142: Pye 2019 and Xavier 2019 are not appropriate references for the identification of auto-oxidation reactions: they are modeling papers. It would be better to cite an early lab study (e.g. Sahetchian et al, Combust. Flame 1991; Crounse & Nielsen. J. Phys. Chem. Lett., 2013)*

The more appropriate citations have been added. The Pye 2019 and Xavier 2019 papers have been acknowledged as modeling papers.

7. *Tables S4, S5: please provide a key to or explanation of the chemical compound code names?*

The key for explanation for the chemical names are found immediately following table S4 (previously table S5) at the end of section S10.

8. *Line 158: Please briefly discuss the conceptual movement of the reaction products between reactivity levels in your 51-box matrix. What does this represent, in a chemical-physical sense? Are there any constraints on the sum of all the dynamic alpha_sub_i values?*

This discussion has been added. Line 155 : "As gas products originating from the precursor HC are atmospherically aged, they become more oxidized, which leads to the addition of functional groups. This atmospheric aging can augment the product distribution, forming more reactive, less volatile, or photolytic products." The sum of the alpha values represent the total amount of oxidation products produced (excluding radicals and species with vapor pressures higher than lumping group 8 in the matrix) in a unit mass per unit mass of precursor consumed. Generally the sum of alpha values is near 1. It can be greater than 1 as the attachment of functional groups can increase the mass of the products.

9. *The following works are cited in the text but are missing from the reference lists: Line 198: Pankow (1994); Line 272: Yap (2011); SI: Roldin (2019)*

These works have been added to the citations.

10. *Line 45: please correct the reference (Robinson et al., 2007)*

This reference has been corrected.

11. *Lines 46-73: Please mention the names of the models used in Hodzic (2010), Cappa (2013), Zhang & Seinfeld (2013)*

The model names have been added to the text

12. *Line 50: Note that, since Lee-Taylor et al used the larger linear alkanes as surrogates for all larger species, the fact that the majority of SOA precursors in their model were linear alkanes is a logical outcome of those SOA species having larger carbon numbers.*

This has been noted in the text.

13. *Lines 61-65: Perhaps "neglect" would be kinder than "fail to consider"? The previous authors are likely well aware of the limitations of their approaches, and (at least in one case) mention those limitations explicitly.*

The language has been updated. This now reads: "Lee-Taylor et al. notes that their reasonable agreement for SOA predictions with field data is not necessarily realistic as they do not include alternative pathways for SOA formation such as particle-phase reactions"

14. *Line 86: "... the typical ozone mechanisms.." This seems vague. Did this study use several ozone mechanisms or just one?*
Only one ozone mechanism was used. This now reads: "In order to increase the feasibility of the model, UNIPAR was also integrated with the carbon bond mechanism (Emery et al., 2015)"

15. *Line 133: "OMAR" needs subscripts.*

Subscripts have been added

16. *Lines 272-274: This sentence is confusing. Please rearrange it?*

This sentence has been rearranged. It now reads: "The value of $K_{w,i}$ is used along with the $k_{on,i}$ to predict intermediate product wall loss using an analytical equation from the study by Han and Jang (2020) as follows"

17. *Line 305: Do you mean "from" instead of "form"?*

Yes. This has been corrected.

18. *Line 321, 323: Do you mean figure 5 instead of fig 6?*

Yes. This has been corrected.

19. *Line 322: Do you mean figure 4 instead of fig 5?*

Yes. This has been corrected.

20. *Line 341: why is it notable that previous experiments used H2O2 as a low-NOx OH source? What are the implications? Please discuss.*

The main difference is that their low $NO_x$ conditions had much lower concentrations of $NO_x$ compared to our experiments. This sentence has been removed and the respective low $NO_x$ concentrations for each of the other studies has been noted within the text.

Line 339: "Zhang and Seinfeld (2013) found that SOA yields for Dodecane (C12) were higher in lower NOx conditions (<6 ppb). Cappa et al. (2013) found significant differences in SOA yields between high and low NOx conditions after 10 hours of oxidation, with higher yields for the low NOx condition (<1 ppb)."

21. *Line 357: please specify whether you mean the relative or absolute contribution of OMAR?*

The relative contribution. This has been specified within the text.

22. *Line 370: do you mean Figure S6 instead of Fig 5?*

    Yes. This figure number has been corrected (Now figure S8 instead of S6).

23. *Line 625, 655: "OCEC is missing its "/""*

    This has been corrected.

**Response to Reviewer 2**

Thank you for reviewing this paper. We hope we have appropriately addressed all of your concerns below.

1. *The use of an outdoor chamber to conduct systematic tests of NOx and particle seed type on SOA yields was a flaw in the experiment design. The day-to-day variability in conditions, namely temperature and light, clouded the very impact of these variables the authors sought to test on SOA yields.*

   While we agree that the day-to-day variability can cause some uncertainty in the experimental data, the use of an outdoor chamber can allow for the testing of the model in ambient-like conditions. Temperature and light are both recorded and are input to the model. Conclusions regarding $NO_x$ and seed impact are drawn from model sensitivity testing rather than the experimental data. The model's ability to predict the experimental data well lends credibility to the conclusions derived from the sensitivity tests.

2. *The explanation of the observed morning SOA burst in some of the experiments that the model fails to capture was seriously lacking*

   A reference to a previous paper which also observed the same phenomenon with another precursor has been added as support to the explanation. Line 329: "A similar trend of rapid SOA formation early in the morning has been observed previously in chamber studies for highly concentrated d-limonene (Yu et al., 2021), which can be rapidly oxidized and form low volatility products."

3. *The discussion on the effect of NOx on autoxidation, the main driver of SOA formation they report, is thin. What is the implemented rate of autoxidation for each alkane precursor? A sensitivity test of the rate of autoxidation on SOA yield is reported on line 370 to be shown in Figure 5, which does not show that at all (only the sensitivity results of wall loss rates). What are the levels of HOx and NOx in the chamber for the duration of each experiment? Both would undoubtedly affect RO2 fates and lifetimes, neither of which is discussed in the section of NOx impact on autoxidation.*

   The implemented rates of autoxidation are the same of each of the precursors. Table S4 shows the reactions and rates present in the autoxidation mechanism for C12. Mechanisms added for C9, C10, and C11 have the analogous reactions for those respective compounds with the same rate

constants. This reference to Figure 5 was inappropriate and has been corrected (Now Fig. S8). Two figures have been added to the SI. The first figure (Fig. S4), which replaced the table in section S7, shows the fraction of autoxidation mass created at several different $NO_x$ values and two different carbon numbers using data from the $NO_x$ sensitivity tests. The second figure (Fig. S5) displays the time profiles of $HO_2$ and $NO_x$ compared to Integrated reaction rates of selected $RO_2$ species. Fig. S5 shows that the autoxidation reaction of the respective $RO_2$ species were insignificant to reactions with NO in both the high and low $NO_x$ conditions. Thus, the addition of autoxidation reactions did not have a significant impact on $RO_2$ fates. Additionally, the autoxidation reaction rate of the $RO_2$ species was somewhat increased in the low $NO_x$ condition compared to the high $NO_x$ condition. This discussion has been added to section 4.4 (Line 343)

4. *Figure 6 shows that SOA yields increase with decreasing NOx. However, Figure 2 shows that the model underestimates SOA loading at high NOx and overestimates at low Nox. This, as a result, possibly undercuts the key model results of SOA yield vs Nox shown in Figure 6. Proper explanation of glaring caveats is not discussed.*

As explained in the "General Notes" section, the initial HC concentrations of the C10 simulations were wrong and have now been updated to the correct values. In addition, the change in the rate constant has improved predictions.

5. *Authors go on to attribute the decrease in SOA yield with Nox to the partitioning of non-autox products, but that explanation is unclear and buried in a Table in the SI. A clearer demonstration with a graphic is needed.*

Fig S4 has been added to improve the explanation.

6. *I would have like to have seen optimization of model parameters (volatility, reactivity, and/or aging) using the suite of observations made. Or at the very least, show Figures 6 and 7 (which are the key figures) but with observations shown as well. This would ground the model and allow the authors to make conclusions that currently seem like extreme extrapolations. For instance, the levels of SOA in the chamber are beyond atmospherically relevant. The high loadings most likely affected the relative contribution of gas partitioning versus heterogeneous chemistry to SOA loading. Additionally, the authors conclude from this modeling exercise that straight chain alkanes are important for urban SOA - without showing much evidence. This is not backed up by the results shown.*

Unfortunately, the addition of observations to Figures 6 and 7 would not be appropriate as those figures are simulated under specific conditions (specific sunlight profile, constant temperature and RH) and relatively low concentrations compared to the outdoor chamber experiments. A sentence has been added to the paper which caveats our conclusions regarding $OM_{AR}$ fraction in relation to particle loadings. Line 359 : "Notably, the levels of SOA loading found in the chamber experiments are much greater than those observed in the ambient environment. Lower SOA loadings may change the relative distribution of $OM_P$ and $OM_{AR}$." An additional reference to a study regarding origins of SOA in industrial cities has been added to support our conclusion that long-chain alkanes are important in urban environments (Line 406).

7. *Line 153: Why 51 species? Where is this from?*

6 levels of volatility and 8 levels of reactivity = 48 species + 3 species lumped explicitly = 51 species

8. *Line 24: symbol in front of C15 did not render*

    This issue has been corrected

9. *Line 125: multiplication symbol did not render properly*
    *This issue has been corrected.*

10. *Line 128: the "i" in alpha_i needs subscript*

    This has been corrected.

11. *Figure S1. Is that starting from an alkoxy radical? Initial compound is blurry. How does the hydroperoxy group convert to a carbonyl? Please explain in the caption.*

    Image has been updated. Figure explanation has been added.

12. *Instead of "or" for organic phase, "in" for inorganic, consider using "org" and "inorg" for clarity*

    This update has been made.

13. *Figure 2. Consider not using black and grey for MCM and MCM+autox. Something more different.*

    Grey line has been replaced with dashed, black line

14. *Line 321. Should cite Fig 5 not Fig 6?*

    This has been corrected.

---

## Editor Decision (ED1)

Previous reviewer comments, amended by editor suggestions:
Reviewer 1, comment 1: Please be more specific how and where references were added and novel aspects are discussed.

Reviewer 1, comment 2: (also Reviewer 2, comment 2): Both reviewers pointed out that the SOA burst in the morning needs more explanation. Referring to previous studies where this has been observed as well is not sufficient.
I restate the reviewer's suggestion to do a sensitivity test and check whether your explanation in Section 4.3 may be valid that a high OH concentration may have caused this peak. Is the required OH concentration even realistic? What other factor could lead to such a peak? Were the conditions in the limonene study by Yu et al comparable to yours?

Reviewer 1, comment 4: I agree with the reviewer that it is very confusing to use identical symbols for different quantities. Please choose different symbols for different parameters.

Reviewer 2, comment 5: How and where was this reviewer comment addressed? Just adding a figure is not sufficient as an explanation.

Reviewer 2, comment 6: The reviewer asked for an optimization of model parameters. They also pointed out that your conclusions on the role of straight-chain alkanes may be misleading. Just adding another reference does not suffice to back up your model-based conclusions.
Please address the reviewer comment "the authors conclude from this modeling exercise that straight chain alkanes are important for urban SOA - without showing much evidence. This is not backed up by the results shown" by justifying in detail your conclusions despite the extreme extrapolations that implies.

Reviewer 2, comment 7: Is this information added in the text?

Additional editor comments:

Scientific:

l. 83: In the abstract, you write that in addition to NOX levels and seed conditions also temperature was controlled. In addition, referee 2 asked how light conditions were monitored. Please be consistent and clear which conditions where controlled and how.

l. 121: It is not clear what you mean by 'The UNIPAR model has been demonstrated...'. What was demonstrated? E.g. good performance to predict SOA formation ? Or do you mean '... has been applied...'?

l. 130: What do you mean by 'mathematically' here? Can it be omitted? If not, please explain.

l. 132: In the abstract, you state that there are three main pathways. Here, you refer to two pathways. Be consistent to avoid confusion.

l. 139: What do you mean by 'The atmospheric process of alkanes'? 'Atmospheric oxidation'?

l. 150/151: Please revise this sentence. It is not clear what exactly is lumped.

l. 158: What are 'photolytic products'? And why do you write 'or'? It seems that products could be more reactive and less volatile and originating from photlysis. Or did I misunderstand what you mean here?

l. 176: Again, the use of 'mathematically' is not clear here. If it is necessary, please explain what you mean here.

Section 4.2: This section is very descriptive and brief though it is a key section of your paper. Please add more discussion on likely reasons of the better agreement in some but not in all cases.

l. 369 – 371: Please be more quantitative here. How much higher are the SOA loadings in the chamber as compared to typical environmental conditions? With your model, you should be able to do a sensitivity study to explore how this might affect the OM(p) and OM(AR) ratio.

Section 4.5: 1) The header of this section is very general, compared to the content. 'Model parameters' may also include temperature, vapor pressures, product distributions etc. However, in fact, you only varied two rate constants. Please clarify this in the section header.
2) What is the reasoning to change the rate constant by +/- 50%. This seems a very narrow range. Please add some references that show that such rate constants indeed only vary within this range.

l. 381: Please add 'of SOA' in this sentence. ('source of SOA')

l. 403-405: Is this a result form your study or by Gentner et al?

Please add a section 'Summary and Conclusions'.

Technical:

l. 13: replace 'formation' by 'prediction' (hen you can omit ''better predict' at the end of the sentence).

l. 14: It is not clear what 'lumping groups' are. Do you mean 'lumped into volatility-reactivity based groups'?

l. 47: replace 'have often' by 'has often'

l. 49/50: the addition of 'using hte CHIMERE regional air quality model' seems at the wrong place. It should be inserted either at the very beginning of the sentence or after 'of SOA'.

l. 53: replace 'predicted' by 'reproduced'

l. 60: which 'current explicit mechanism' do you mean here? Are PRAM not included in any available mechanism or specific in MCM?

l. 72: add 's' to precursor

l. 73: add 'for' (account for…)

l. 124: replace 'mechanisms' by 'mechanism'

l. 175: MCM stands for 'Master Chemical Mechanism'; thus, MCM mechanisms seems redundant.

l. 177: replace 'increase' by 'increases'

l. 215: replace 'or' and 'in' by 'org' and 'inorg'. Please check carefully the full manuscript for other instances (eg.. Eq-6, 7, 10, 11 , l. 227, l. 233 etc).

l. 221: Is there anything missing or should the colon at the end of the sentence be replaced by a period?

l. 344: replace 'lowly volatile' by 'low volatility'

---

## Author Response (AR2)

**Previous reviewer comments, amended by editor suggestions:**

*Reviewer 1, comment 1: Please be more specific how and where references were added and novel aspects are discussed.*

**Response**. References were added as follows:

Section 3 (Model Description): The first sentence of this section refers to previous works in which the UNIPAR model has been demonstrated for other precursors. The sentence immediately following establishes that this study's novel aspect is the extension to linear alkanes.

Section 3.2 (Lumping and aging): The first sentence of this section establishes that the dynamic lumping array structure as well as the incorporation of aging was developed in previous studies. The sentence immediately following establishes that alkane products were lumped in this study according to that previous structure. The last sentence of this references a previous paper in which some of the information regarding the lumping structure and aging within this section was discussed in more detail.

Section 3.3: This section is entirely novel to this study and thus, is described with no references to previous development of the model.

Section 3.4 and 3.5: These sections outline equations determining the partitioning and particle-phase reactions present within the model. Each equation has appropriate references to previous works in which these equations were developed.

Section 3.6: The second sentences of this section explains that the method used in this section was developed in previous studies,

*Reviewer 1, comment 2: (also Reviewer 2, comment 2): Both reviewers pointed out that the SOA burst in the morning needs more explanation. Referring to previous studies where this has been observed as well is not sufficient. I restate the reviewer's suggestion to do a sensitivity test and check whether your explanation in Section 4.3 may be valid that a high OH concentration may have caused this peak. Is the required OH concentration even realistic? What other factors could lead to such a peak? Were the conditions in the limonene study by Yu et al comparable to yours?*

**Response**.  In order to response to the comment from the reviewer and the editor, the additional chamber experiment has been conducted for C13 with the lower hydrocarbon (35ppb) and HONO (35ppb) concentrations that those in Table 1.  The spike in SOA mass was significantly reduced and SOA mass was better predicted with the UNIPAR alkane model.

Similar to alkane of this study, the fast reaction of d-limonene with oxidants at low NOx levels can yield the rapid SOA growth with low volatility products.  If d-limonene SOA forms with the low concentration of hydrocarbon, the spike in SOA mass will not occur.

The newly added sentence can be found in the end of Sect. 4.3 and reads now,

"When both the concentrations of HONO and the long-chain alkane are reduced, this spike can be suppressed as seen in Figure S3"

*Reviewer 1, comment 4: I agree with the reviewer that it is very confusing to use identical symbols for different quantities. Please choose different symbols for different parameters.*

**Response**: The symbol for polarizability has been changed to $P_i$ rather than $\alpha_i$.

*Reviewer 2, comment 5(What are the levels of Hox and Nox in the chamber for the duration of each experiment? Both would undoubtedly affect RO2 fates and lifetimes, neither of which is discussed in the section of Nox impact on autoxidation. Authors go on to attribute the decrease in SOA yield with Nox to the partitioning of non-autox products, but that explanation is unclear and buried in a Table in the SI. A clearer demonstration with a graphic is needed.): How and where was this reviewer comment addressed? Just adding a figure is not sufficient as an explanation.*

**Response**: Figure S4 was added to the SI as a clearer demonstration of the impact of alkane autoxidation products on yield differences between high and low NOx conditions, as well as different carbon number precursors, as demonstrated by a model sensitivity test. This replaced a previous table in which the fraction of autoxidation mass was reported for various high and low NOx experimental simulations, which was relatively difficult to interpret. The new Figure S4, however, is a straightforward figure which supports our assertion that "The fraction of autoxidation products to the total SOA mass generally increases with decreasing carbon number or decreasing $NO_x$ level due to gas-particle partitioning of non-autoxidation lowly volatile products…". Additionally, figure S5 was added which shows the HOx and NOx concentrations from the sensitivity test, as well as the integrated reaction rates of selected RO2 species. This contributes to the added discussion of RO2 fates :

"Additionally, the impact of the autoxidation reactions on RO2 chemistry is examined using the Integrated Reaction Rates in Fig S6. Generally, the autoxidation reactions of selected RO2 species displayed (HO3C106O2, and HO3C126O2 from the MCM mechanisms of C10 and C12, respectively) were insignificant compared to reactions with NO. Additionally, the autoxidation reaction rate was slightly increased under low NOx conditions compared to high NOx conditions, which is consistent with Fig. S5."

*Reviewer 2, comment 6: The reviewer asked for an optimization of model parameters. They also pointed out that your conclusions on the role of straight-chain alkanes may be misleading. Just adding another reference does not suffice to back up your model-based conclusions. Please address the reviewer comment "the authors conclude from this modeling exercise that straight chain alkanes are important for urban SOA - without showing much evidence. This is not backed up by the results shown" by justifying in detail your conclusions despite the extreme extrapolations that implies.*

**Response**: The determination of the model parameters was performed by using the solver in the spreadsheet and are thus already optimized.

In order to response to the reviewer's comment, we modified our claim which now reads:

"Thus, we conclude that long-chain alkanes are an important source of SOA formation in air parcels originating from diesel fuel combustions."

*Reviewer 2, comment 7(Line 153: Why 51 species? Where is this from?): Is this information added in the text?*

**Response**: Yes, this information is in the text : "The $\alpha_i$ array consists of 6 different reactivity levels (very fast, fast, medium, slow, partitioning only, and multi-alcohol) and 8 different volatility levels based on vapor pressure which represent 48 species, along with 3 explicit species that are lumped separately (glyoxal, methylglyoxal, and epoxydiols)."

Additional editor comments:

**Scientific:**

*l. 83: In the abstract, you write that in addition to NOX levels and seed conditions also temperature was controlled. In addition, referee 2 asked how light conditions were monitored. Please be consistent and clear which conditions where controlled and how.*

**Response**: Temperature conditions were not controlled within the experiment. It is clear how this can be confusing when temperature is mentioned alongside NOx and seed conditions and thus, "temperature" has been removed from that line in the abstract. The information on how temperatures, RH, and light conditions were monitored can be found within the "Experimental section":

"A hygrometer (CR1000 measurement and control system, Campbell Scientific Inc., USA) was used to measure meteorological factors (temperature, relative humidity (RH) and an ultraviolet radiometer (TUVR, Eppley Laboratory Inc., USA) was used to measure sunlight intensity"

*l. 121: It is not clear what you mean by 'The UNIPAR model has been demonstrated...'. What was demonstrated? E.g. good performance to predict SOA formation ? Or do you mean '... has been applied...'?*

**Response**: This sentence has been updated: "The UNIPAR model's ability to  accurately simulate SOA formation from various aromatic HCs (Im et al., 2014; Zhou et al., 2019; Han and Jang, 2022a), monoterpenes (Yu et al., 2021), and isoprene (Beardsley and Jang, 2016) has been previously demonstrated."

*l. 130: What do you mean by 'mathematically' here? Can it be omitted? If not, please explain.*

**Response**: The word "mathematically" has been removed.

*l. 132: In the abstract, you state that there are three main pathways. Here, you refer to two pathways. Be consistent to avoid confusion.*

**Response**: This sentence has been updated and now reads: "The SOA mass in the model forms via three pathways: OM produced via multiphase partitioning of organic products ($OM_P$),aerosol phase reactions of organic species to form $OM_{AR}$ via oligomerization in the *org* phase, and reactions in the wet *inorg* phase which also form $OM_{AR}$ (acid-catalyzed oligomerization and organosulfate (OS) formation)."

*l. 139: What do you mean by 'The atmospheric process of alkanes? 'Atmospheric oxidation'?*

**Response**: Yes. "atmospheric process" has been replaced with "atmospheric oxidation."

l. 150/151: Please revise this sentence. It is not clear what exactly is lumped.

**Response**: This sentence has been divided into two sentences to improve clarity and now reads:

"The lumping structure of the UNIPAR model, along with a dynamic $\alpha_i$ array which considers aging, has been developed in previous studies (Zhou et al., 2019; Han and Jang, 2020; Yu et al., 2021). According to this structure, the alkane oxidation products, originating from simulations at various $NO_x$ levels (HC ppbC/$NO_x$ ppb = 2~50) using the MCM and alkyl peroxy radical autoxidation mechanisms, were lumped ."

*l. 158: What are 'photolytic products'? And why do you write 'or'? It seems that products could be more reactive and less volatile and originating from photolysis. Or did I misunderstand what you mean here?*

**Response**: Photolytic products are products produced via photolysis by "photolysis products" may be clearer language. Generally atmospheric aging will result in products which are more reactive and less volatile. The general exception is when compounds undergo photolysis in the atmosphere which will create products which are more volatile, and possibly more reactive.

This sentence has been updated for clarity and now reads: "This atmospheric aging can augment the product distribution, forming more reactive and less volatile products via oxidation, or photolysis products which are more volatile but may be more reactive."

*l. 176: Again, the use of 'mathematically' is not clear here. If it is necessary, please explain what you mean here.*

**Response**: The word "mathematically" has been removed from this sentence.

*Section 4.2: This section is very descriptive and brief though it is a key section of your paper. Please add more discussion on likely reasons of the better agreement in some but not in all cases.*

**Response**: Additional discussion has been added to the end of this section:

"As discussed in sect. 3.3, Eq. 4  for the IVC was fitted to the product distributions of C10, C11, and C12 using unified coefficients a and b to account for the vapor pressure drop in products that results from the increase of the carbon number of the precursor. The vapor pressure drop is not linear as the addition of a carbon to a longer carbon chain causes a smaller decrease in volatility (i.e. the reduction of volatility from adding a carbon to C11 will be smaller compared to the reduction in volatility from adding a carbon to C10). Eq. (4) is non-linear but  may not be the ideal equation to model the vapor pressure drop within the product array. Thus, as a consequence of the fitting process, the model performance of the C10 array may have been sacrificed to improve model performance in the case C11, and C12.  Furthermore, the current explicit gas mechanism cannot include every possible product, which can be associated with the oxidation

on all carbons in each alkane chain. The products of this study are obtained from oxidation at a given carbon position, which is representative of the whole set of gas oxidation. This can cause some deviation of prediction from the true atmospheric oxidation process. Thus, the product distribution originating from these explicit mechanisms can also cause variation in prediction of SOA mass."

*l. 369 – 371: Please be more quantitative here. How much higher are the SOA loadings in the chamber as compared to typical environmental conditions? With your model, you should be able to do a sensitivity study to explore how this might affect the OM(p) and OM(AR) ratio.*

**Response**: The sensitivity test for the $OM_p$ to $OM_{ar}$ was performed at atmospherically relevant SOA loadings ($OM_0 = 5$ µg/m$^3$). The note that SOA loadings found in the chamber were much greater than those observed in the ambient environment was meant as a caveat that the validation of model performance was under relatively high SOA loadings. It is clear that our sentence that read : "Lower SOA loadings may change the relative distribution of $OM_P$ and $OM_{AR}$." suggested that the sensitivity test itself was performed under conditions that were not atmospherically relevant and has thus been removed.

These sentences have been updated and now read:

"Fig. 7 also includes the contribution of OMP and OMAR to SOA in the absence of inorganic seed at atmospherically relevant conditions (OM0 = 5 µg/m3). Overall, the relative contribution of OMAR is higher with lower temperatures and higher carbon numbers. Notably, the levels of SOA loading found in the chamber experiments which were used to validate model performance are much greater than those observed in the ambient environment (up to 32 times higher than the annual PM air quality standard set by the EPA of 12 µg/m3 (Epa, 2012))."

Section 4.5:

1) The header of this section is very general, compared to the content. 'Model parameters' may also include temperature, vapor pressures, product distributions etc. However, in fact, you only varied two rate constants. Please clarify this in the section header.

**Response:** The heading of this section was changed to "Uncertainty of model rate constants."

*2) What is the reasoning to change the rate constant by +/- 50%. This seems a very narrow range. Please add some references that show that such rate constants indeed only vary within this range.*

**Response:** Previous studies have reported that calculated rate constants for autoxidation reactions have uncertainties of about a factor of 5. This uncertainty test has been updated such that this rate is increased/decreased by a factor of 5 and this information has been updated in section 4.5.

*l. 381: Please add 'of SOA' in this sentence. ('source of SOA')*

**Response:** This correction has been made.

*l. 403-405: Is this a result form your study or by Gentner et al?*

**Response:** This is a result from our study. This result is compared to the result from Gentner et al.'s study later on: "Notably, Gentner et al. (2012) found that the peak of SOA mass production from linear alkanes in diesel appears between C19 and C22 which is a slightly higher carbon number compared to our prediction."

*Please add a section 'Summary and Conclusions'.*

**Response:** The atmospheric implication section has been divided into two sections : "4.6: Application of IVC-base product distributions to SOA simulation from diesel-linear alkanes" and "4.7 Summary and Conclusions."

Technical:

*l. 13: replace 'formation' by 'prediction' (hence you can omit ''better predict' at the end of the sentence).*

**Response:** The sentence has been updated and now reads: "Autoxidation paths integrated with alkyl peroxy radicals were added to the Master Chemical Mechanismv3.3.1 to improve the prediction of low volatility products in the gas phase and SOA mass"

*l. 14: It is not clear what 'lumping groups' are. Do you mean 'lumped into volatility-reactivity based groups'?*

**Response:** Yes. This sentence has been updated and now reads: "The resulting gas products were then lumped into volatility-reactivity based groups that are linked to mass-based stoichiometric coefficients."

*l. 47: replace 'have often' by 'has often'*

**Response:** "have often" has been replaced by "has often"

*l. 49/50: the addition of 'using the CHIMERE regional air quality model' seems at the wrong place. It should be inserted either at the very beginning of the sentence or after 'of SOA'.*

**Response:** This sentence has been updated and now reads: "Using the CHIMERE regional air quality model, Hodzic et al. (2010) found that the inclusion of IVOCs led to a substantial improvement in predictions of SOA compared to data collected from the MILAGRO field experiment"

*l. 53: replace 'predicted' by 'reproduced'*

**Response:** "predicted" has been replaced with "reproduced"

*l. 60: which 'current explicit mechanism' do you mean here? Are PRAM not included in any available mechanism or specific in MCM?*

**Response:** This is referencing the MCM mechanism. "current explicit mechanism" has been changed to "current MCM mechanism".

*l. 72: add 's' to precursor*

**Response:** This correction has been made.

*l. 73: add 'for' (account for...)*

**Response:** This correction has been made.

*l. 124: replace 'mechanisms' by 'mechanism'*

**Response:** This correction has been made.

*l. 175: MCM stands for 'Master Chemical Mechanism'; thus, MCM mechanisms seems redundant.*

**Response:** "Mechanisms" has been removed from this phrase.

*l. 177: replace 'increase' by 'increases'*

**Response:** This correction has been made

*l. 215: replace 'or' and 'in' by 'org' and 'inorg'. Please check carefully the full manuscript for other instances (eg.. Eq-6, 7, 10, 11 , l. 227, l. 233 etc).*

**Response:** These corrections have been made.

*l. 221: Is there anything missing or should the colon at the end of the sentence be replaced by a period?*

**Response:** This colon has been replaced by a period.

*l. 344: replace 'lowly volatile' by 'low volatility'*

**Response:** This correction has been made.